# Variation in Water-Holding Capacity in *Sphagnum* Species Depends on Both Plant and Colony Structure

**DOI:** 10.3390/plants13081061

**Published:** 2024-04-09

**Authors:** Willem Q. M. van de Koot, James Msonda, Olga P. Olver, John H. Doonan, Candida Nibau

**Affiliations:** 1National Plant Phenomics Centre, Institute of Biological, Environmental and Rural Sciences, Aberystwyth University, Aberystwyth SY23 3EE, UK; willem.vd.koot@hotmail.com (W.Q.M.v.d.K.); olo8@aber.ac.uk (O.P.O.); 2Department of Computer Science, Llandinam Building, Aberystwyth University, Aberystwyth SY23 3DL, UK; jam139@aber.ac.uk

**Keywords:** moss, desiccation, peatlands, pore space, µCT

## Abstract

Peatlands have become a focal point in climate mitigation strategies as these ecosystems have significant carbon sequestration capacities when healthy but release CO_2_ and other greenhouse gases when damaged. However, as drought episodes become more frequent and prolonged, organisms key to the functioning of some peatlands are increasingly under pressure from desiccation. The *Sphagnum* mosses, which tend to keep their ecosystem waterlogged and many of whom promote peat formation, are only mildly desiccation-tolerant in comparison to other mosses. The role of *Sphagnum* anatomy and colony structure is poorly understood in the context of desiccation resilience. Using four different *Sphagnum* species belonging to four different subgenera and positions along the gradient of the water table, we show that plant morphological traits and colony density are important determinants of water storage capacity. Our results show that, as previously postulated, the majority of the water is stored in an easily exchangeable form, probably extracellularly, and that plant morphological traits, specifically the type and presence of branches, are major contributors to water storage and can explain some of the interspecies variation. We also show that plant density is another important determinant for water storage capacity as higher densities hold larger quantities of water per unit of biomass for all four species, which increases resilience to desiccation. The results presented here suggest that species choice and planting density should receive more attention when considering peatland restoration strategies.

## 1. Introduction

Peatlands are estimated to account for approximately 25% of all soil-sequestered carbon globally [1,2], making them one of the largest terrestrial carbon stores. One of the main threats to the functioning of peatlands, whether bogs or fenland, is reduced ground water. When peat becomes desiccated, it switches from being a carbon sink to a source, as the accumulated dead organic matter is exposed to oxygen and releases greenhouse gases [3]. Mosses of the genus *Sphagnum* L. have a pivotal role in bog ecosystems; through their large water storage capacity and acidifying properties, they can promote the accumulation of organic matter, aiding peat formation and ultimately sequestering carbon for geologically significant time periods.

The increasing prevalence of drought affecting peatlands can be both directly and indirectly attributed to anthropogenic activity [4]. Lack of precipitation for prolonged periods is becoming more common globally as a consequence of climate change, and drainage of peatlands for agriculture, horticulture or fuel has damaged the water table regulatory capacity of many of these ecosystems. 

Many bryophytes can tolerate desiccation, coupling metabolic activity to water availability and recovering quickly from dry episodes. However, *Sphagnum* species are largely desiccation-intolerant and thus have strict preference for wet or very humid habitats. *Sphagnum* mosses have retained at least some of the molecular responses to drought that are found in other bryophytes [5,6], but they are unable to fully revive after severe desiccation, suggesting the loss of some physiological functions required to survive drying. Instead, they have evolved characteristics that manipulate their local environment in order to reduce the likelihood of lethal desiccation [7,8,9,10,11,12,13,14]. Therefore, *Sphagnum* mosses have been labelled ‘desiccation avoiders’. The effect of desiccation on *Sphagnum* is of particular interest in the context of habitat restoration, as *Sphagnum* may be re-introduced into damaged and unstable ecosystems where intermittent desiccation is highly probable [9,12,15]. Water table regulation in such peatlands depends, at least in part, on direct water retention by *Sphagnum* mosses [10]. They also have indirect effects as the environment they contribute to creating and maintaining becomes highly unfavourable to many other plant species with higher water use that would deplete the soil water reservoirs [16].

*Sphagnum* plants have multiple adaptations thought to aid water retention. Firstly, specialised water storage hyaline cells [17], make up around 80% of the plant volume [10,16]. These dead and empty hyaline cells are found on all *Sphagnum* leaflets as well as on the outside of the stems. Moreover, the presence of small pores in the walls of hyaline cells may contribute to water conductivity along the stem [14]. Secondly, *Sphagnum* plants have two types of branches, known as ‘spreading’ and ‘pendent’ branches, that may aid in the uptake and transport of water through *Sphagnum* colonies. The spreading branches extend laterally from the main stem, potentially aiding water capture from precipitation as well as water exchange between individual plants in a colony [18]. Pendent branches, appressed downwards along the stem, may wick water upwards from the water table to the meristematic areas at the top of the stem in the capitulum, the growing head of the plant [13,19,20]. All these different components form a complex and connected system of spaces where water could be transported and stored. It has been estimated that up to 90% of all water retention is extracellular, stored in the capillary spaces formed between the branches [10,21], allowing individual *Sphagnum* to hold over 2000% their own biomass in water [5,9,10,22].

Plant density, a potential third mechanism for water retention, has been poorly investigated. *Sphagnum* mosses typically grow in dense colonies with visually distinctive and diverse growth forms that have been described as, lawns, mats, hummocks and tussocks. The numerical density of plants within colonies might be affected by inherent genetic factors specific to each species such as size of the capitulum and stem branching. However, microclimatic conditions of the growth site could also affect variation [11]. Thus, in drier habitats more dense colony formations might have an advantage as this would reduce the area of individual plants exposed to airflow by reducing surface roughness and evaporation [18]. If this is the case, then plant density should be less important in wetter habitats such as pools and pond margins where water loss can easily be replenished. In other bryophytes, colony structure and morphology was found be associated with differences in drying rates [23], and increased plant density was correlated with increased water retention capacity across species [11]. 

The *Sphagnum* genus, therefore, may deploy different water-holding mechanisms to avoid desiccation and we hypothesise that the explanatory factors are, at least in part, due to variation in plant and colony morphology. However, how these vary across species and habitats as well as the contribution of each one to the whole remains poorly studied. Here we investigate the contribution of the different water storage mechanisms using four *Sphagnum* species, *S. fallax*, *S. capillifolium*, *S. papillosum* and *S. inundatum* (representing the four subgenera, namely *Acutifolia*, *Cuspidata*, *Sphagnum* and *Subsecunda*). These species were collected along of a gradient of water table positions across a single natural mire habitat. To specifically examine water-holding capacity, plants were maintained under identical conditions immediately prior to and during experiments, thus removing the effect of differences in ground water or any other microclimatic field conditions such as shading, community composition, surface slope, evaporation rate and surface roughness. We show that saturated water content was not correlated with dry weight, suggesting that the majority of the water is stored extracellularly. We experimentally test the contribution of branches to water storage and found that they make a significant contribution to the amount of water stored and can explain some of the interspecies variation. Finally, we tested the effect of plant density on desiccation. Increased plant density was found consistently to improve desiccation resilience in all four species, as higher densities had a larger absolute water-holding capacity, but this was not directly correlated with increased biomass. Therefore, we demonstrate roles for both plant and colony architecture in explaining the variation in water-holding capacity between species.

## 2. Results

### 2.1. Sphagnum Water Content Is Not Correlated with Plant Dry Weight

We have previously shown that different *Sphagnum* species collected from a single mire had different water-holding capabilities that did not relate to their position along the water table [5,24]. When we compared the saturated Relative Water Content (RWC) of each species with the corresponding dry weight, we did not observe a correlation between the two (Figure 1). This observation raised the possibility that a significant part of the water content is stored in one or more extracellular compartments, and that the contribution of these compartments may vary between species.

We reasoned that part or all of the extracellular water would be loosely bound; therefore, this exchangeable water could be estimated by a simple approach such as mild centrifugation [25]. To quantify the amount of exchangeable water, individual plants for each of the four *Sphagnum* species were cut to the same height, allowed to fully saturate with water and weighed (saturated weight). Then, each individual plant was placed in a centrifuge tube (Appendix A), spun at increasing speeds with the amount of water recovered by spinning weighed (extracted water weight) and used to calculate the water lost at each speed. The plants were allowed to fully dry and weighed again (dry weight) and this value was used to calculate the Relative Water Content (RWC) for each plant. 

All species lost increasing amounts of water when the speed was increased from 17 g to 66 g and to 149 g (Figure 2A). At higher centrifugation speeds (266 g), little extra water was lost suggesting that most of the easily extractable water is removed by speeds of 149 g (Figure 2A). *S. papillosum* lost water more quickly, while *S. capillifolium* lost the least water. *S. inundatum* and *S. fallax* showed intermediate rates of water loss. For saturated RWC, the opposite was observed. *S. capillifolium* has the lowest water content and lost the least amount of water while *S. papillosum* showed the highest RWC (Figure 2B) and the highest amount of water loss, suggesting that in *S. papillosum* more of the water can be removed by gentle centrifugation while *S. capillifolium* has the lowest levels of easily exchangeable water. We suggest that these observations can be related to the anatomy of each species. While *S. papillosum* has long and large branches, *S. capillifolium* plants are slender with shorter branches, so have a smaller surface area to store water. The post centrifugation RWC was the same for all species (Figure 2B) suggesting that the easily exchangeable water was indeed extracellular. These suggest water storage capacity per unit biomass is unaffected, but the absolute water storage capacity is strongly affected.

### 2.2. The Role of Plant Architecture in Water Storage

Likely extracellular water storage sites include the spaces between branches and leaflets and the gaps within a colony between neighbouring plants. We therefore tested the effect of branch removal on easily exchangeable water storage of individual plants using the centrifugation procedure as described above (Appendix A). Due to the postulated functional differences between spreading and pendent branches, we removed each separately and then together and measured the amount of water recovered by centrifugation (Figure 3). Removal either of spreading branches or of pendent branches led to less recovered water compared to intact plants, indicating that both types of branches contribute to water storage. While the amount of recovered water was affected by branch removal in all species, *S. capillifolium* showed the smallest effect (Figure 3). When all branches were removed, only very small amounts of water were recovered, supporting the notion that branches are important stores of easily exchangeable water (Figure 3). Consistent with the results shown in Figure 1, we did not find differences in water content per gram of plant material between the species when the different types of branches are removed (Appendix A). On the other hand, the presence of branches affected the absolute water weight (g), again suggesting most of the water is stored extracellularly (Appendix A), with *S. fallax* being the exception.

Since we hypothesised that the centrifugally labile water is held in spaces around the plants, we subjected isolated plants to μCT scanning (Figure 4A) and calculated the volume of each plant as well as the external surface area (Figure 4B). As expected from visual observations, *S. papillosum* showed the largest volume of plant material and the largest surface area, and *S. fallax* the lowest (Figure 4B). Differences were smaller between the other two species, indicating that the architecture of individual plants is only partially responsible for the differential water content of the four species. 

### 2.3. The Effect of Colony Density on Water Storage 

The results above suggest that the architecture of individual plants can only partially explain the differences in water retention between *Sphagnum* species. Under natural conditions, the four *Sphagnum* species form colonies at different stem densities (1.9 plants cm^−2^ for *S. capillifolium*, 1.65 plants cm^−2^ for *S. fallax*, 1.2 plants cm^−2^ for *S. papillosum*, 0.6 plants cm^−2^ for *S. inundatum* [5,24]. *S. capillifolium* grows furthest from the water table and has the higher stem densities while *S. inundatum*, a pool species, shows the lowest plant density [5]. Analysis of moss cores images by µCT scanning confirmed this (Figure 5). *S. capillifolium* moss cores have the highest volume of plant material while the pool species *S. inundatum* the lowest. In *S. papillosum*, while the absolute volume of plant material is lower than for *S. capillifolium*, the branched plant structure provides an increased surface area (Figure 5). The use µCT scanning allowed us to calculate the porosity of the moss cores (void volume/total volume) and the total pore volume, providing a measure of the number of spaces available between plants for water storage. Porosity was higher in *S. inundatum* and lower in *S. capillifolium* (although there is a large variation between *S. capillifolium* cores, Figure 5) as expected from the former being a pond species and the later a hummock species. Despite having a lower capacity for water storage per plant, *S. capillifolium* is found in dense cushions with tightly packed plants creating small pores which likely hold on to water more tightly. 

To experimentally test the effect of colony density on water-holding capacity, we artificially manipulated colony densities by arranging the four *Sphagnum* species at various densities in cosms [5] and subjected them to a three-week drought period followed by one week of rewetting to test for physiological recovery. Over the course of the experiment, the cosms containing the *Sphagnum* plants were weighed twice weekly and these weights were used to calculate water content for each time point. Photosynthetic capacity as expressed by the *Fv/Fm* ratio of chlorophyll fluorescence was also measured, providing a measure of the physiological effects of water loss. 

For all species under drought, water content was found to reduce relatively steadily over time. Increased plant density significantly increased the time it took for water content to decrease (Figure 6). *S. papillosum* and *S. inundatum* consistently started with higher water content at the different densities, and lost water more slowly while *S. fallax* and *S. capillifolium* lost water more quickly (Figure 6). Higher stem densities increased the colony’s ability to retain water across time for all species. *S. inundatum* at 40 stems per cosm never reached the low water levels seen at lower stem densities (Figure 6). Although all species showed slower water loss rates with increased density, this was more apparent for *S. inundatum* and *S. papillosum* (Figure 5). All species returned to their pre-drought condition after rewatering.

Water loss is expected to affect numerous physiological processes, including photosynthesis. *Fv/Fm* measurements, which provide a dynamic estimate of photosynthetic ability, broadly confirm the gravimetric results (Appendix A). For all species at the 10-plant density, *Fv/Fm* sharply decreased, and after 1 week of drought exposure, the cosms containing either *S. fallax* and *S. capillifolium* lost nearly their entire photosynthetic capabilities (*Fv/Fm*~0.200) while *S. inundatum* and *S. papillosum* appeared to be more resistant at this density (Appendix A). All species showed photosynthetic recovery after re-watering, with better recovery at higher stem densities, and this was more substantial for *S. inundatum* and *S. papillosum*. Since the higher density colonies spent less time at low water content, they may have suffered less damage to their photosystems.

Taken together, the results indicate that both stem density and plant architecture contribute to increased water holding, which can result in a prolonged ability to withstand desiccation. 

## 3. Discussion

The unique biology of *Sphagnum* plays a central role in the ecology of and carbon sequestration in bogs. Fundamental to this role and, indeed, to the stability of the entire bog ecosystem, is the mosses’ ability to retain large amounts of water. *Sphagnum* lacks many of the hydration controls of higher plants, such as stomata and thick cuticles, so water management is primarily via uptake and storage. As with other land plants, water uptake depends on hydraulic connectivity with ground water reservoirs. Unlike vascular plants, that have a well-defined vascular system and thus a discrete range of pore sizes, the pore size and connectivity in *Sphagnum* are more complex as water is conducted through spaces of variable dimensions, some of which can also act as storage [26,27]. 

The structural basis of variation in water storage has been based on largely correlative studies across species [9]. Here, we experimentally manipulate aspects of plant architecture and plant density to quantify their roles in the water storage capabilities of four different *Sphagnum* species that occupy different positions along the water table. To deconvolve the effect of water transport and water storage, we used conditioned ex situ material. This allowed us to study water storage independent of the species hydraulic connectivity with water reserves in the lower peat. We confirm that most of the water is stored in a readily exchangeable form consistent with the majority being held in the pore spaces within and between plants [10,21], and experimentally demonstrate that architectural features such as specialised branch structures as well as the density and packing of stems do indeed contribute to water retention. Experimental manipulation of these features suggests that different adaptations have been differentially adopted by the four species. 

### 3.1. The Role of Colony Density 

*Sphagnum* rarely grows as isolated plants and typically form complex (and sometimes multi-species) colonial communities. Although the role of the *Sphagnum* colony structure has not previously been experimentally investigated, plant density has been recognized as a useful descriptive trait to separate hummock and hollow assemblages [9,11,12,28,29]. On the other hand, for a given species, colony density can change between different habitats, and environmental variation is suggested to be a more likely factor in determining colony density than species or genetic differences [30]. μCT imaging of moss cores (Figure 5) shows that the increased plant density provides not only more branches to store water but also affects the pore spaces between stems, where water can also be held. This approach holds promise for a thorough quantitative understanding and modelling of the relationships between moss structure and water dynamics at the plant level, a scale which has hitherto been difficult to measure.

Despite having similar observed morphologies and plant volume, *S. inundatum* is found in shallow pools where it has direct access to surface water throughout most of the year, while *S. papillosum* is found higher above the water table and may be more reliant on precipitation. *S. papillosum* has a higher natural plant density that, according to our results, should increase its water-holding capacity. *S. inundatum*, having permanent water access, should not be under strong selection pressure for higher plant density. Environmentally correlated variations in density have been previously documented for a number of bryophytes including *Sphagnum* [11] and, in some bryophyte species, density was shown to associated with improved desiccation resilience [23]. *S. capillifolium* had the highest density and is also the species growing highest above the water table at the collection site, forming dense hummocks. Although high plant density can in part be explained by the fact that *S. capillifolium* is a relatively small-sized species, its hummocks are known to be very dense which may allow this species to occupy its niche above the water table by increasing its water-holding capacity by increasing its hydraulic connectivity and reducing water loss [14,26].

Contrary to the traditional view that only hummock species store large quantities of water as drought avoidance strategy [10], all four species can store large quantities of water when manipulated into a dense hummock-type colony arrangement. For all four species, higher plant densities could hold larger amounts of water per unit of biomass (as seen in the increased RWC at higher densities) and maintain this water for longer when subjected to an induced drought treatment (Figure 6). 

Despite the potential impact of colony density for peatland projects, especially projects in which *Sphagnum* is planted on damaged peat surfaces for the purpose of recolonisation, the importance and consequences of this trait are poorly understood. Little experimental data and field data have been collected, as colony density estimation is a labour-intensive task. Recent developments of tools for automated high throughput quantification may enable easier collection of density data [24].

### 3.2. The Role of Plant Morphology

Plant morphology, particularly the spreading and pendent branches, also plays an important role in water-holding capacity. The pendent branches are appressed along the main stem and are presumed to function as a wick, allowing water to move up towards the capitulum [10,13,19,20,31]. The spreading branches extend laterally with the potential to interact with neighbouring plants, creating voids that may be a suitable size for water storage and affect water movement from below. The four species used here show quite different morphologies in terms of the number and length of pendent and spreading branches that could contribute to their different water-holding capacities. Both *S. papillosum* and *S. inundatum* are larger with dense lateral branches and long pendent branches (Figure 4) and most of the centrifugally removable water was present in the spreading and pendant branches (Figure 3). On the other hand, *S. capillifolium* plants are slender with sparse short branches and they lose the least water by centrifugation. There is little difference between the before and after centrifugation RWC, demonstrating that in the case of *S. capillifolium* plant structure contributes less to water retention and colony density may be more important in this case. *S. fallax* shows an intermediate morphology and has intermediate water storage capacity. This is different from what was observed in the colony density experiments (Figure 6) where these two species perform poorly. Although it is difficult to directly compare the two experiments as, in one case, we have single plants where we draw out the water by centrifugation and in the other, different density arrangements of plants, it may be the case that *S. fallax* and *S. capillifolium*, which are also the smallest species, can hold water physically as well than the larger *S. papillosum* and *S. inundatum*, but suffer in fixed colony-size experiments as their smaller size forms a less dense colony in comparison with the two larger species. Interestingly, after centrifugation, there was no significant difference between the RWC of any of the species, further supporting the idea that extracellular water is responsible for most of variation in water-holding capacity between species. Clymo et al. [10] suggested that in *Sphagnum* plants 90% of the water was stored extracellularly. In our experiments, even after centrifugation the plants retain at least 40% of their weight. This could be due to the water stored in the hyaline cells which could also be considered extracellular water but not easily removed by centrifugation. Water in these compartments is subjected the large surface tension and capillary force [14,27], potentially making it difficult to remove by gentle centrifugation. 

Besides the importance of the different types of branches as shown here, leaf curvature has also been correlated with water-holding capacity [9] and in both cases enables the creation of external spaces that can hold water tightly and do not drain easily [14]. Habitat preference modelling also indicate that capitulum size is an important predictor for the species position along the water table and thus may also be related to desiccation tolerance [32]. We find that this is true for *S. papillosum* (large capitulum, high position along the water table), but *S. capillifolium* is a hummock species with small capitula. These and other studies highlight the varied and interdependent strategies that members of the *Sphagnum* species utilise to avoid desiccation. 

In summary, the choice of species may be important as they differ in desiccation resilience, but this can be partly overcome by increased inoculation densities. Optimising *Sphagnum* colony composition and density may significantly improve plant survival and establishment in peatland restoration projects.

## 4. Materials and Methods

### 4.1. Plant Material

Four species of *Sphagnum S. fallax*, *S. papillosum*, *S. capillifolium*, *S. inundatum*, representing the subgenera *Acutifolia*, *Cuspidata*, *Sphagnum* and *Subsecunda*, were collected from a blanket bog at the foot of Pen y Garn (SN791758), which is mostly ombrotrophic (as described in [5,24]). Single-species-dominated sods of approximately 650 cm^2^ were cut out whilst retaining the colony structure, and transferred into rectangular boxes (Gratnells Ltd., Harlow, UK, inner size 28.5 × 23.5 × 14.5 cm) with 9 drainage holes in their bottom. Water table distances in each area were measured across four consecutive seasons and were 11.4 ± 4.2 cm for *S. fallax*, 16.4 ± 3.1 cm for *S. papillosum*, 17 ± 3.5 cm for *S. capillifolium* and *S. inundatum* was collected from a pool. The mosses were kept outside on the south-east facing side of a glasshouse on Plas Gogerddan in shallow trays with approximately 3 cm of standing rainwater, ensuring the *Sphagnum* had direct uniform access to water. The moss was allowed to acclimatise for at least two weeks prior to the experiments, and were kept for the duration of the experiments, of which repeats were conducted across multiple seasons.

### 4.2. Water Removal by Centrifugation

Ten individual representative plants per species were selected, cut to a uniform length (5 cm) and centrifuged using a Thermo Fisher Scientific Heraeus Megafuge 16R (Waltham, MA, USA) with a TX-200 swinging bucket rotor. Each plant was placed in a 15 mL tube that was set up to allow the released water to collected separate from the plant (see Appendix A for a graphic representation of this set up). Generally, each plant was centrifuged for 5 min at each speed of 17 g, 66 g, 149 g and 266 g at 4 °C. Each plant was weighed using a Sartorius M-power precision balance AZ124 before the first centrifugation and after the final centrifugation, and again once they had dried for at least 24 h at 80 °C and were at constant weight. After each centrifugation episode, the tubes were removed and held with their base on a flat surface so a line could be drawn to mark the meniscus of the water that had been released. The water was then weighed and recorded according to the interval at which it came out. This method was repeated with plants that had been dissected to remove either the spreading branches, the pendent branches, or both (Appendix A). The weight of the water released at each speed was used to calculate the percentage weight loss after each centrifugation interval, based on the initial weight before centrifugation. The relative water content (RWC) was calculated as the difference between either the before- or the after-centrifugation weight and the dry weight, divided by the dry weight, expressed as a percentage relative to the weight before centrifugation.

### 4.3. Micro Computed Tomography (µCT) Scanning and Image Analysis

Twenty-four *Sphagnum* plants representative for each of the four species (Figure 4) were cut to the same height (7 cm) and placed in a 14 mm diameter µCT holder and scanned in a Scanco µCT100 scanner (Scanco Medical, Wangen-Brüttisellen, Switzerland) at a resolution of 29.6 voxel. The moss cores were taken from well-established moss cushions using the 73 mm diameter µCT scanner holder. They were fully hydrated before scanning at a resolution of 73.6 voxel. The resulting image stacks (in a proprietary ISQ format, Scanco Medical, Switzerland) were imported into 3D Slicer version 5.2.2. 3D Slicer [33] is an open-source application software for 3D image processing and visualisation. A segment editor module within the application was then used for thresholding and removal of unwanted components. Thereafter, volume and surface area statistics were extracted from the segmented 3D models. Due to the presence of water at the base of the cores, only the top 4 cm of each moss plant and moss core were used for analysis. To obtain pore space volume and porosity, open-source Python image analysis libraries, OpenCV (4.7.0) and Insight Toolkit (ITK) version 5.3.0 were used. Pore space refers to the volume of the void space or gaps within the moss core and between the core and its convex hull. Porosity is the fraction of the volume of void space to the total volume within the convex hull. In simple terms, a convex hull of a moss core is the smallest convex shape that encompasses a given set of voxel points representing the core.

### 4.4. Colony Density Experiments

Individual representative plants of each of the four species, as described above, were arranged into microcosms at 10, 20, 30 or 40 plants, cut to a uniform size (5 cm) and placed in a vertical microcosm consisting of a square Petri dish, of which one side was removed to allow for evaporation [5]. The cross-sectional area available for the plants and evaporation was 20 cm^2^, meaning the density in the cosms was 0.5, 1, 1.5 or 2 plants cm^−2^. However, in this text the densities will be referred to as 10, 20, 30 and 40, after the respective number of plants in each cosm. Six cosms per density per species were prepared, of which two were allocated to the control group and the remaining four to the experimental group. The microcosms were randomised and kept in a shallow tray containing 1 cm of standing water for two weeks prior to the experiment under temperature of 22–24 °C, relative humidity between 40–60% and light 90–110 µmol m^−2^ s^−1^. All cosms in the experimental group were subjected to a three-week drought period by transfer to a dry tray, followed by one week of recovery when they were transferred to a tray containing 1 cm of standing water as before, whilst the control group was kept standing in 1 cm water for the duration of the experiment. Cosms were weighed twice weekly during the drought period and RWC was calculated as (wet weight-dry weight)/dry weight) × 100%. The efficiency of photosynthesis was assessed three times weekly through measurement of the chlorophyll fluorescence parameter *Fv/Fm* in a CF Imager (Technologica, Essex, UK) following the manufacturers protocol, which provides an estimate of the efficiency of photosystem II.

### 4.5. Statistical Analysis

Statistical analyses were carried out in RStudio version 4.0.3 [34] using the packages car [35] and ggpubr [36]. Data visualisation and plotting was conducted in Python, with the packages Seaborn [37] and Matplotlib [38]. Statistics that involve the dry weight measurements were limited to data collected on the final day of the corresponding experiment, as this was the day after which the plants were dried and measured. When data were normally distributed, as indicated by a normality test (Shapiro–Wilk) after the removal of outliers, parametric *t*-tests were used to calculate significance of difference between normally distributed data comparisons, and Pearson product moment correlation was used to test for correlation between variables. When data were non-parametric, a Wilcoxon Rank-Sum test was used to test significance of difference between sample groups.

## Figures and Tables

**Figure 1 plants-13-01061-f001:**
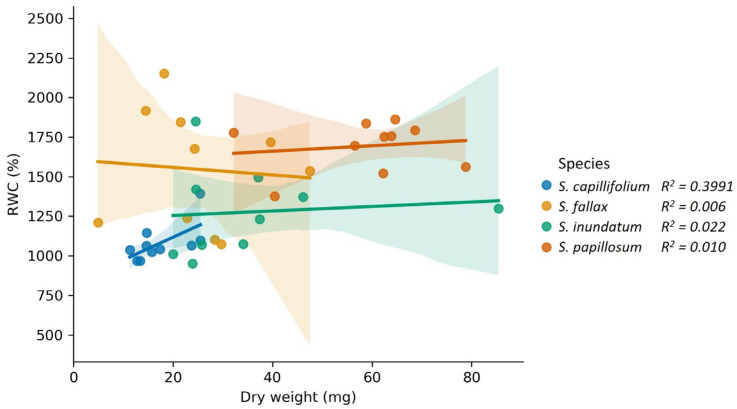
Correlation between dry weight and Relative Water Content (RWC) for four different *Sphagnum* species, namely *S. fallax*, *S. papillosum*, *S. capillifolium*, *S. inundatum*, as indicated. Each data point represents an individual plant, lines represent the mean and the shaded areas the 95% confidence interval for each sample. R^2^ values are shown for each species.

**Figure 2 plants-13-01061-f002:**
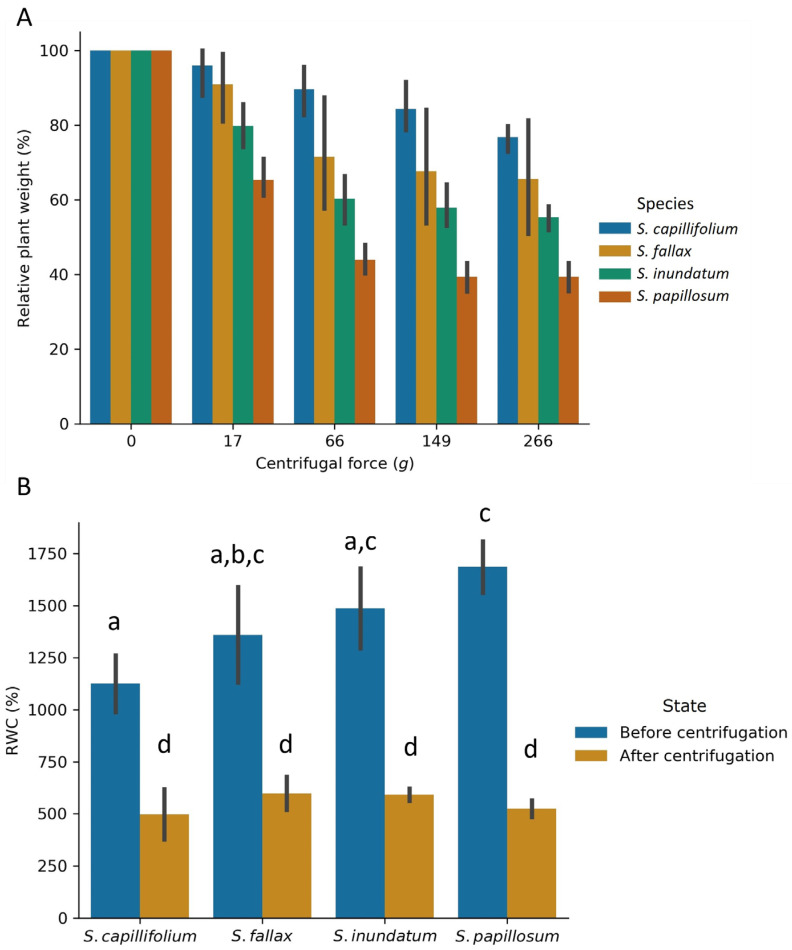
(**A**) Percentage weight loss of the four *Sphagnum* species at different centrifugal forces as indicated. Plants were centrifuged at the different centrifugal speeds as indicated for 5 min and the weight of the water released at each step used to calculate the percentage weight loss. (**B**) The RWC before centrifugation and after the final centrifugation was determined. Statistical analysis can be found in Appendix A. The data show average ±SD for 10 plants. See Appendix A for details of the experimental set up. Letters above bars represent significantly different values for *p* < 0.05.

**Figure 3 plants-13-01061-f003:**
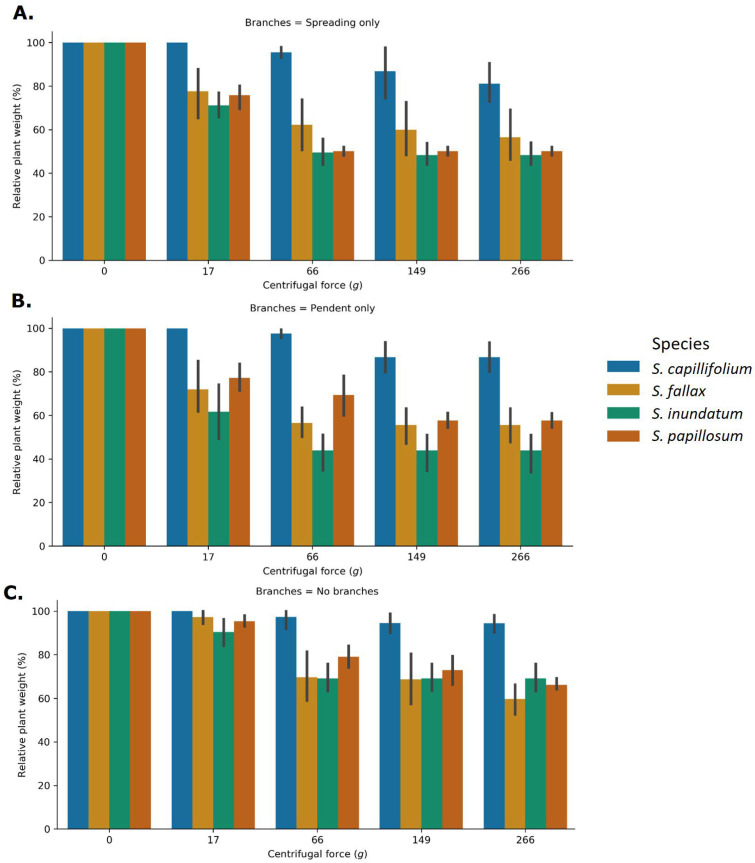
Effect of branch removal on water storage capacity in the different *Sphagnum* species. Single *Sphagnum* plants for the four species as indicated were used. Pendent branches (**A**), spreading branches (**B**) or both branch types (**C**) were selectively removed (Appendix A) and the plants centrifuged as described in Methods. Percentage weight loss was calculated for each centrifugal force. Data represent average ± SD of 10 plants. Statistical analysis can be found in Appendix A.

**Figure 4 plants-13-01061-f004:**
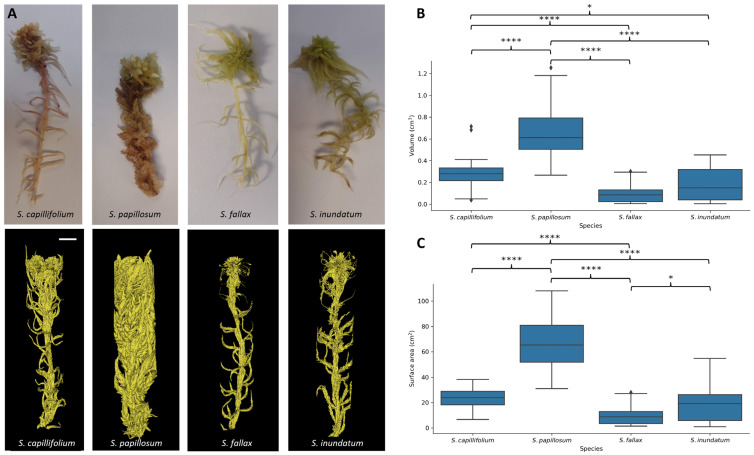
(**A**) Single plant images representing the four *Sphagnum* species used in this study as indicated. Top row RGB images, bottom row 3D rendered images obtained by μCT scanning. (**B**,**C**) Volume and surface area of individual plants as determined by µCT scanning. Single *Sphagnum* plants of comparable lengths were scanned, the volume (sum of voxels containing plant material) (**B**) and the surface area (sum of external pixels) (**C**) was calculated from segmented images. Data show averages and interquartile ranges for 24 plants per species. * indicates that values are significantly different for *p* < 0.05 and **** for *p* < 0.0001.

**Figure 5 plants-13-01061-f005:**
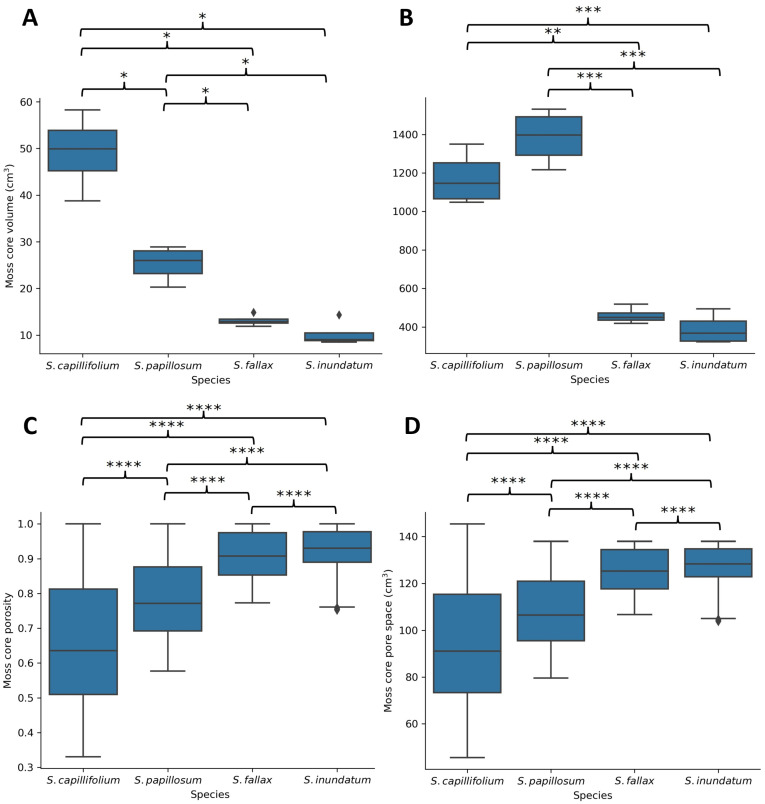
Volume (**A**), surface area (**B**), porosity (**C**), and pore volume (**D**) of moss colonies of the four different *Sphagnum* species as determined by µCT scanning. Four moss cores (7 cm diameter) were scanned and volume (sum of voxels containing plant material) surface area (sum of external pixels), porosity (void volume/total volume) and total pore volume was calculated from segmented images. Data represent average and the interquartile range for four moss cores per species. Asterisks indicate significance levels * = *p* < 0.05, ** = *p* < 0.01, *** = *p* < 0.001 and **** = *p* < 0.0001.

**Figure 6 plants-13-01061-f006:**
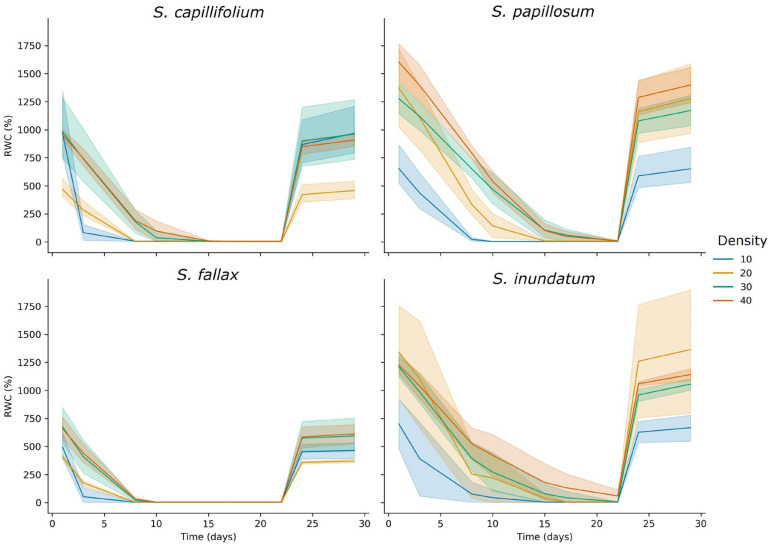
Effect of manipulated plant density on RWC during an imposed drought scenario. Plants in cosms at the indicated stem density for each species were subjected to water withdrawal for 21 days after which they were re-watered. Water content was calculated twice weekly during the experiment. Lines represent the mean and the shaded areas the 95% confidence interval for each sample.

## Data Availability

All data supporting the findings of this study are available within the paper and within its Appendix A published online.

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
