# Peer review of "Variation in Water-Holding Capacity in Sphagnum Species Depends on Both Plant and Colony Structure"

_plants, 2024, doi:10.3390/plants13081061_

Round 1

Reviewer 1 Report

Comments and Suggestions for Authors

·     make sure to always write scientific names in italics (e.g. Lines 197, 219, 355, 396 f.)

·     Line 30: Do not repeat words from the title in the keywords (e.g. Sphagnum, colony)

·     Site conditions, in particular water table, for the sampling plots would be helpful for a better explanation of the results of the experiments, but they are missing.

·     The capitula are of different sizes in the selected species and can also vary within a species. The capitulum was not removed in the experiments, but presumably has a different influence on the results, also depending on the length of the mosses. This was not sufficiently taken into account in the manuscript and is one major point of criticism.

·     Check the allocation of individual text passages to the chapters, shorten the introduction and discussion

Introduction:

The reader gets the impression that Sphagnum forms all peatlands. Please distinguish between bogs and fens, when you want to emphasise the role of the genus Sphagnum. The world’s peatlands are only partly dominated by Sphagnum. Additionally, only some of the Sphagnum species are key species and accumulate peat. Of the species selected for the experiments, only S. capillifolium and S. papillosum are peat-accumulating. Please be more careful in the presentation of the investigation context and in the interpretation. Additionally, in bogs you have to consider the bog water table, which in an intact (raised) bog is above the groundwater table (line 39/40). In natural habitats Sphagnum can withstand short periods of drying out well, this is one of the self-regulating mechanisms of a (raised) bog. But with long-lasting desiccation of Sphagnum the resulting death of the mosses leads, in particular, to the collapse of the self-regulating system of (raised) bogs and to the stop of C accumulation or even to CO2 emissions and related reduction of water retention capacity. The clearer differentiation between natural/ intact and drained/ degraded peatlands and restoration of peatlands seems appropriate, and the effects on Sphagnum.

·     Sphagnum species grow in typical ecological niches, with the water level as one of the most important factors. However, the moss morphology within one species can differ (even in one mire) (see Couwenberg et al. 2022, doi:10.1002/ecs2.4031) also because of different light and nutrient conditions. This effect is not (clearly) taken into account in the introduction.

·     Paragraph Line 103 ff.: this is more of an abstract than an introduction

·     Line 107 f.: it is unclear, whether the single species were only collected in one place each (with a specific water table) or each species was collected at different water table positions across a single natural mire habitat.

·     Line 109: which other microclimatic field conditions do you mean?

·     Clear hypotheses are missing in the introduction

Methods:

·     Line 396: name the species also here

·     Line 397: the site conditions remains unclear but are crucial for understanding. Please add information, in particular the (mean) water table of the sampling plots.

·     Line 398: it is unclear, whether the sods consist of only one (pure) species or several species

·     Line 399: the colour of the boxes is not important ;)

·     Line 402: the length of the mosses is unclear, thus also the water table below moss surface. How often and how was the water added to ensure the standing water of 3 cm? What was the quality of the water?

·     Line 407: did you use mosses with similar morphology for this experiment? The length of the mosses and therefore the proportion (and effect) of the capitulum is unclear. Additionally, 10 plants per species are not very much and it is unclear, whether the variability within one species is sufficiently mapped.

·     Line 413 f.: It is very difficult to measure the weight of single dried peatmosses, because they immediately absorb the humidity in the air and constantly changes their weight. Please provide more information on weight conditions, scale, accuracy of the measured values etc.

·     Line 424: use the spoken name of µCT in the subheading

·     Line 425: Were the 24 plants per species took in their natural compound or “reassembled”?

·     Line 435: how did you ensure that the top 4 cm of the moss plants were without (extracellular) water?

·     Line 445: which uniform size (length)?

·     Line 453: concretise the controlled environment in the experiment: light, air humidity, water table

·     Line 456: how the mosses were watered (from above or below)?

·     Line 457 f.: Was there condensation on the cosms and how was this taken into account (for determination of the weight)?

Results:

·     The methods are partly described in this chapter, but relatively imprecisely, which would be superfluous if the methods chapter were placed before the results chapter.

·     Fig. 1: results relate to dry weight of individual moss plants. The length of the mosses and the proportion of capitula in the total weight are unclear, which is important for the interpretation of the results.

·     Hypotheses should be part of the introduction

·     Parts of this chapter are already interpretation/ discussion.

·     Fig. 2, 3: colours could be choosen barrier-free (e.g. red green weakness) à e.g. white, light grey, dark grey, black or with hatching/ shading

·     Fig. 3 B, C: labelling (numbers) on the x-axis is missing 

·     Fig. 4: pictures and graphs should be always in the same order, at least in one figure

·     Line 207-209: What is the size of the capitula of each species? This probably is one main reason for different stem densities.

·     Line 218-220: please check the assignment of hummock and pond to the named Sphagnum species

·     Line 245: you probably mean water content, not water table

·     Fig. 6: for S. fallax and S. capillifolium you show 0% RWC after 10 or 15 days, respectively. This seems to be somewhat contradictory to the results in Fig. 2a, where these two Sphagnum species hold the water best. How can this be explained? (in the discussion).

Discussion:

·     This chapter contains large parts of introduction and repetition of results. Shorten the discussion and structure it better (possibly with subheadings).

·     Line 273: how does Sphagnum occurrence lead to paludification?

·     Line 298: next to colony density also moss morphology can change between different habitats

·     Line 310: the morphology of the branch leaves is not similar. Be more precise.

·     Line 319 f.: you should add data on site conditions (esp. water table) at the sampling plots to be able to understand different positions above the water table

·     The Sphagnum species were selected on the basis of their “natural” different positions above the water table (written in the abstract), but in the discussion you describe S. papillosum and S. capillifolium as hummock, and S. inundatum and S. fallax as hollow species. S. pap and S. fal are known as lawn/ carpet species (see Laine et al. 2018, Daniel & Eddy 1985, Rydin & Jeglum 20).

·     Line 354 f.: please check the reference

·     Line 358: water stored in hyaline cells is considered to be intracellular

·     364 f.: What influence does the water-holding capacity of the leaves (“does not drain easily”) have on CT measurements?

·     Line 367 ff.: the conclusion is not comprehensible, because in Fig. 1 you show the water content of individual plants, not only of capitula.

·     Line 371: This (previous comment) also means that the conclusion “habitat preference is not strong for this mire” is not comprehensible. Additionally, S. pap. is more a lawn than a hummock species. The presentation of data on site conditions (in particular water table) would clarify the relationship.

·     Line 373 f.: be careful with the conclusion on the role as peatland ecosystem engineers (see above)

·     Line 393 (and others): the planting of plugs is only common in bog restoration projects in UK, please do not generalise. If good water availability can be ensured, there are other, far less costly methods of Sphagnum application.

Author Response

Thank you for your careful revision of our manuscript and the valuable comments provided. We have endeavoured to address each comment as best as we could and give details of the changes after each comment and also in the main manuscript as tracked changes.

Comments and Suggestions for Authors

  • make sure to always write scientific names in italics (e.g. Lines 197, 219, 355, 396 f.)

Thank you. We have now corrected this.

  • Line 30: Do not repeat words from the title in the keywords (e.g. Sphagnum, colony)

Keywords changed to reflect this.

  • Site conditions, in particular water table, for the sampling plots would be helpful for a better explanation of the results of the experiments, but they are missing.

Thank you for your suggestion. Although we have previously published detailed descriptions of the site and water table and refer to them (Nibau et al., 2022 and van de Koot et al., 2021), we agree that, for ease of interpretation it would be useful to have the values in this manuscript. We have included them in the methods section and clearly refer the reader to our earlier paper for detailed description. 

  • The capitula are of different sizes in the selected species and can also vary within a species. The capitulum was not removed in the experiments, but presumably has a different influence on the results, also depending on the length of the mosses. This was not sufficiently taken into account in the manuscript and is one major point of criticism.

Thank you for the comment and a very good point as the size of the capitulum will have an influence on the results as part of the general morphology of the plant. We now make this clearer in lines 90 and 380. As for the length, in each experiment (except the microCT colony scanning) the mosses were cut to the same size and we now make this clear in the methods section.  For the centrifugation experiments, removal of the capitulum led to structural instability in the remaining material and difficulty in handling the material thus we did not evaluate with and without capitulum.

  • Check the allocation of individual text passages to the chapters, shorten the introduction and discussion

As requested, we have made the requested changes in the introduction and removed material that is not necessary to understand the data presented. We have also re-structured the discussion by dividing it into sections.

Introduction:

The reader gets the impression that Sphagnum forms all peatlands. Please distinguish between bogs and fens, when you want to emphasise the role of the genus Sphagnum. The world’s peatlands are only partly dominated by Sphagnum.

Thank you – we make this clearer in the abstract and explicitly mention the main types of peatlands in the first part of the introduction.

Additionally, only some of the Sphagnum species are key species and accumulate peat. Of the species selected for the experiments, only S. capillifolium and S. papillosum are peat-accumulating. Please be more careful in the presentation of the investigation context and in the interpretation. Additionally, in bogs you have to consider the bog water table, which in an intact (raised) bog is above the groundwater table (line 39/40). In natural habitats Sphagnum can withstand short periods of drying out well, this is one of the self-regulating mechanisms of a (raised) bog. But with long-lasting desiccation of Sphagnum the resulting death of the mosses leads, in particular, to the collapse of the self-regulating system of (raised) bogs and to the stop of C accumulation or even to CO2 emissions and related reduction of water retention capacity. The clearer differentiation between natural/ intact and drained/ degraded peatlands and restoration of peatlands seems appropriate, and the effects on Sphagnum.

Thank you for this insightful comment. While different species of Sphagnum do indeed show different rates of decomposition and litter formation, several studies suggest that the colony morphology, water availability and climatic factors are more important than the species and under favourable conditions all species will decompose (see for example 10.1007/s11104-018-3579-8, https://doi.org/10.1002/ece3.2119). We feel that this distinction is not directly relevant to our manuscript as we do not investigate decomposition or peat formation and this would further lengthen the introduction and discussion. However, to avoid implying that all species are equally important for peat formation in all environments, we have changed the abstract section to ‘The Sphagnum mosses, which tend to keep their ecosystem waterlogged and many of whom promote peat formation, are only mildly desiccation tolerant in comparison to other mosses.’

In relation to the water table, all our experiments were done with acclimatised material that has uninterrupted access to water. But we agree that we need to be cautious when generalizing to the field and have revised the discussion to reflect this.

  • Sphagnum species grow in typical ecological niches, with the water level as one of the most important factors. However, the moss morphology within one species can differ (even in one mire) (see Couwenberg et al. 2022, doi:10.1002/ecs2.4031) also because of different light and nutrient conditions. This effect is not (clearly) taken into account in the introduction.

We do agree that species morphology is affected by the environment and now make this more clear in the introduction and when we discuss factors affecting water holding capacity.

  Paragraph Line 103 ff.: this is more of an abstract than an introduction.  Clear hypotheses are missing in the introduction

We have reworded lines 102-109 to provide an explicit hypothesis and re-worded lines 103f.

  • Line 107 f.: it is unclear, whether the single species were only collected in one place each (with a specific water table) or each species was collected at different water table positions across a single natural mire habitat.

All the species were collected from within a 200m2 area at water table positions that we now include in the methods. In addition, and as mentioned in line 107 ‘plants were maintained under identical conditions immediately prior to and during experiments, thus removing the effect of differences in ground water.’ 

  • Line 109: which other microclimatic field conditions do you mean?

We now expand on the type of microclimate conditions that can affect the moss in line 110.

Methods:

  • Line 396: name the species also here

Species names now included.

  • Line 397: the site conditions remains unclear but are crucial for understanding. Please add information, in particular the (mean) water table of the sampling plots.

As mentioned above, we described the site conditions in detail in our other publications. For clarity we now include the mean water table for where the different species were collected here as well as more explicit reference to details in our previous work.

  • Line 398: it is unclear, whether the sods consist of only one (pure) species or several species

We harvested the sods to contain mostly a single species. In some cases there were a few plants of other species but the majority in each case was single species. We now include this information in the methods.

  • Line 399: the colour of the boxes is not important ;)

Agreed, not critically important in current context. It was important for other experiments and papers involving the same material (where the boxes were used for image analysis).

  • Line 402: the length of the mosses is unclear, thus also the water table below moss surface. How often and how was the water added to ensure the standing water of 3 cm? What was the quality of the water?

Thank you. The mosses kept in boxes were of different lengths depending on the depth of the sod we could harvest. When we harvested the sods we made sure to collect deep enough to reach the dead anaerobic area of the moss and this varies from species to species. For the majority of the experiments (except for the moss colony CT scanning) the plants were separated and all cut to the same length. We now include the specific length in the methods section. As described, the boxes had holes at the bottom and were kept in large trays. Rainwater collected at the site was added to the trays during dry weather as required to maintain approximately 3cm of standing water.

  • Line 407: did you use mosses with similar morphology for this experiment? The length of the mosses and therefore the proportion (and effect) of the capitulum is unclear. Additionally, 10 plants per species are not very much and it is unclear, whether the variability within one species is sufficiently mapped.

Thank you. We used representative plants from each species and they were all cut to the same length. We agree that having more plants is always better to determine intra-specific variation but this is a time consuming experiment where the speed of preparation was paramount to have reproducible results. To mitigate the low number of plants per experiment, we independently repeated the experiment several times with comparable results.

We have clarified the methods section as follows:

‘Ten individual representative plants per species were selected, cut to a uniform length (7 cm) and  centrifuged using…’

  • Line 413 f.: It is very difficult to measure the weight of single dried peatmosses, because they immediately absorb the humidity in the air and constantly changes their weight. Please provide more information on weight conditions, scale, accuracy of the measured values etc.

We were aware of the tendency of moss to absorb humidity and indeed measured its rate as part of other experiments.  This varied on species and the tissue but was between 0.1 and 0.3% per minute under typical open lab conditions for an intact 4cm plant.  We therefore took all feasible steps to ensure this did not have a serious impact on the results and interpretation. The moss was dried to constant weight at 80C (at least 2 days) and then transferred to sealed containers and weighed using a Sartorius M-power precision balance AZ124 (Repeatability: +/-0.0002 g, Linearity: +/- 0.0003 g).  We had previously validated the protocol to ensure that we could weigh a single plant to constant weight with reasonable accuracy.

  • Line 424: use the spoken name of µCT in the subheading

Thank you – we have now spelled out the abbreviation in full.

  • Line 425: Were the 24 plants per species took in their natural compound or “reassembled”?

These were scanned as single plants. Re-worded in the methods to make this clear.

  • Line 435: how did you ensure that the top 4 cm of the moss plants were without (extracellular) water?

We obtained cores directly from the moss sods taking care to maintain colony integrity. This necessarily meant that a variable amount of dead and waterlogged matter at the base of the core was also transferred to the scanning tube. Within that waterlogged volume, segmentation of the plant material from the water was impossible so we arbitrarily took it out of the analysis. This indeed left a variable length of plant material, and in order to standardise between samples, we used the upper 4cm of the core, which always provided good segmentation between the wet plant material and air spaces or pores, providing a robust comparison between species.

  • Line 445: which uniform size (length)?
  • Line 453: concretise the controlled environment in the experiment: light, air humidity, water table

Both the length and detailed environmental conditions were added to the relevant methods section.

  • Line 456: how the mosses were watered (from above or below)?

The mosses in this experiment and other experiments were watered from below. We have now re-worded to text to reflect this.

  • Line 457 f.: Was there condensation on the cosms and how was this taken into account (for determination of the weight)?

Significant condensation can occur when closed cosms are moved between different thermal environments so we took several steps to avoid this.  The cosms were open at the top, kept well-spaced and maintained in the same environment during the experiment, so we did not observe any condensation on the cosms.  

Results:

  • The methods are partly described in this chapter, but relatively imprecisely, which would be superfluous if the methods chapter were placed before the results chapter.

The placement of the Methods after the Results is a journal formatting requirement, not under our control. We have included brief description of the methods in the results to help the reader better understand the experiment without the need to always check the methods section.

  • Fig. 1: results relate to dry weight of individual moss plants. The length of the mosses and the proportion of capitula in the total weight are unclear, which is important for the interpretation of the results.

The results of figure 1 are from individual plants cut to a uniform length of 7cm. We have also data for single capitula collected from greenhouse conditions and from capitula harvested in the field. In all cases there is no correlation between dry weight and water content.

  • Hypotheses should be part of the introduction

We have removed this sentence from the results and placed an over-arching hypothesis at the end of the introduction.

  • Parts of this chapter are already interpretation/ discussion.

For each section of the Results we summarise the main results and explore their meaning in that specific context, then in the Discussion we placed this work in a more general context and how it fits in the current knowledge and published literature. We hope it enhances the accessibility to the general reader. In addition, the guidelines for this specific journal states

  • Results: Provide a concise and precise description of the experimental results, their interpretation as well as the experimental conclusions that can be drawn.

  • Fig. 2, 3: colours could be choosen barrier-free (e.g. red green weakness) à e.g. white, light grey, dark grey, black or with hatching/ shading

Thank you for your suggestion. All figures have been re-plotted using the 'Colorblind palette' for Seaborn in Python.

  • Fig. 3 B, C: labelling (numbers) on the x-axis is missing

Corrected. 

  • Fig. 4: pictures and graphs should be always in the same order, at least in one figure

Thank you. This was an oversight from our part, and we have now corrected it.

  • Line 207-209: What is the size of the capitula of each species? This probably is one main reason for different stem densities.

Absolutely, simple space constraints imposed from the capitulum could indeed affect plant spacing.  Equally likely, plant spacing might constrain capitulum size.  For example, S. inundatum has the lowest stem density and one of the largest capitula. Since the direction of causality remains poorly defined (undefined to our knowledge), and the genetic tools to dissect this are surprisingly still absent for these species, we undertook a “surgical” approach that experimentally manipulated stem density. 

  • Line 218-220: please check the assignment of hummock and pond to the named Sphagnum species

It was an oversight from us, now corrected.

  • Line 245: you probably mean water content, not water table

We refer to water content in the mentioned line.

  • Fig. 6: for S. fallax and S. capillifolium you show 0% RWC after 10 or 15 days, respectively. This seems to be somewhat contradictory to the results in Fig. 2a, where these two Sphagnum species hold the water best. How can this be explained? (in the discussion).

Thank you for your insightful comment – this result does seem at variance with Figure 2 and we discuss it more thoroughly in the discussion lines 358f. Figure 2 explores the water retention physics as centrifugal force is used to extract the water from the cells. This is very different from the experimental situation in figure 6, where water loss is caused by evaporation, driven by temperature and ambient relative humidity. Additionally, in the experiment pertaining to figure 2 uses single plants whereas the colony density experiment uses colonies at various densities. The amount of variables differing between these experimental conditions makes the results hard to compare, but it may be the case that S. fallax and S. capillifolium, which are also the smallest species, can hold water physically as well as the larger S. papillosum and S. inundatum, but suffer in fixed colony-size experiments as their smaller size forms a less dense colony in comparison with the two larger species .

Discussion:

  • This chapter contains large parts of introduction and repetition of results. Shorten the discussion and structure it better (possibly with subheadings).

As requested, we have shortened the discussion and included subheadings to improve structure and readability.

  • Line 273: how does Sphagnum occurrence lead to paludification?

We have removed this phrase.

  • Line 298: next to colony density also moss morphology can change between different habitats

Absolutely, but in this paragraph we are discussing colony density in particular. The effect of the habitat on plant morphology has been included in the introduction as requested earlier.

  • Line 310: the morphology of the branch leaves is not similar. Be more precise.

Thank you. We have re-written this sentence to specifically compare similar plant volume drastically different plant densities.

  • Line 319 f.: you should add data on site conditions (esp. water table) at the sampling plots to be able to understand different positions above the water table

Also as requested above, we have now added the average water table for the collection point for each species.

  • The Sphagnum species were selected on the basis of their “natural” different positions above the water table (written in the abstract), but in the discussion you describe S. papillosum and S. capillifolium as hummock, and S. inundatum and S. fallax as hollow species. S. pap and S. fal are known as lawn/ carpet species (see Laine et al. 2018, Daniel & Eddy 1985, Rydin & Jeglum 20).

Thank you for your comment, we agree we were too general in our classifications and have revised the text to reflect the fact that only S. capillifolium is a true hummock species in this locality.

  • Line 354 f.: please check the reference

We believe this reference is correct.

  • Line 358: water stored in hyaline cells is considered to be intracellular

The hyaline cells are dead and, generally, have open pores to the external environment, thus they can be considered intracellular in senso stricto, but in another sense there is no living barrier between their water and the external water. The physical nature of the cells, however, could present a functional restraint to free exchange, based on capillary forces.  Despite this and as it would be very difficult to ascertain the nature of the water in these cells, we have revised the manuscript and where appropriate replaced the term ‘extracellular’ with the phrase ‘easily exchangeable’.

  • 364 f.: What influence does the water-holding capacity of the leaves (“does not drain easily”) have on CT measurements?

The studies mentioned in this paragraph relating to leaf shape were done by other groups and did not involve the use of CT scanning. In Bengtson et al, 2020 the authors use mesocosms experiments to manipulate water level drawdown and then measured many different morphological traits. Using mixed models they found that leaf curvature was an important factor in water holding capacity. In our CT scanning experiments, we are measuring total volume of plant materail and total amount of speces in between this plant material. At the resolution have available, it is not easily possible to measure the effect of leaf curvature.

  • Line 367 ff.: the conclusion is not comprehensible, because in Fig. 1 you show the water content of individual plants, not only of capitula.

Thank you for drawing out attention to this. Indeed in figure 1 we used whole plants for the RWC correlation. We have also measured capitula RWC and found that it also does not show a correlation to dry weight. But as we do not show this data we have reworded this statement.

  • Line 371: This (previous comment) also means that the conclusion “habitat preference is not strong for this mire” is not comprehensible. Additionally, S. pap. is more a lawn than a hummock species. The presentation of data on site conditions (in particular water table) would clarify the relationship.

As mentioned above we have now included the water table measurements and refine our definitions of hummock and hollow species.

  • Line 373 f.: be careful with the conclusion on the role as peatland ecosystem engineers (see above)

We have now removed the mention to their role as ecosystem engineers.

  • Line 393 (and others): the planting of plugs is only common in bog restoration projects in UK, please do not generalise. If good water availability can be ensured, there are other, far less costly methods of Sphagnum application.

Thank you for highlighting this, we have removed the reference to moss plugs and talk only about general peatland restoration as many of the same considerations apply.

Reviewer 2 Report

Comments and Suggestions for Authors

The authors conducted a comarison study on water holding capacity among four peat mosses. The topic is old but of importance for our understanding Sphagnum adaptation to water availabity along the gradient of water table level. I have one major comments as follows:

The authors used centrifuge method to discard water. This method has ever been use by many studies such as Mulligan and Gignac 2001 and Bu et al. 2013.  My concern is how the authors know all the water centriguged out is from outside of the cells. Do not forget, the water could go through the hyline cells from the pores. So, I will recommend the authors to use other terms but not extracellar water although I like to believe most or at least some of them are extracellar. 

I do not have many minor comments but several as below:

Line14, CO2 should be corrected;

Line 17, Sphagnum mosses are ok;

Line 20, subgenera but not genera;

Line 20 and 21, along the gradient of ... would be fine;

Line 211, I am not sure which term is more suitable between moss core and moss monolith?

Line 244, it would be better if the unit for a, b and d can be changed as cm3, cm2 and cm3; 

Line 498 to 581, the scientific name for plants should be italic.  

Author Response

Thank you for revising our manuscript and providing useful comments. We have addressed each comment separately below and changes made are highlighted in the main manuscript by the track changes feature.

The authors conducted a comarison study on water holding capacity among four peat mosses. The topic is old but of importance for our understanding Sphagnum adaptation to water availabity along the gradient of water table level. I have one major comments as follows:

The authors used centrifuge method to discard water. This method has ever been use by many studies such as Mulligan and Gignac 2001 and Bu et al. 2013.  My concern is how the authors know all the water centriguged out is from outside of the cells. Do not forget, the water could go through the hyline cells from the pores. So, I will recommend the authors to use other terms but not extracellar water although I like to believe most or at least some of them are extracellar. 

Thank you for your thoughtful comments. The question is the hyaline cells are part of the extracellular reserve of water was also raised by reviewer 1 that thinks they should be considered intracellular. As we said in our reply to reviewer1, although in hyaline cells water is kept inside the cells so could be considered intracellular, they have open pores to the outside the constitute providing strong water connectivity to the outside (McCarthy and Price, 2014) and in a sense can be considered part of the ‘extracellular’ exchangeable water. Despite this and as it would be very difficult to ascertain, we have revised the manuscript and where appropriate replaced the term “extracellular” and with the phrase “easily exchangeable”.

We have also included the reference of Mulligan and Gignac 2001 as an example for the centrifugation experiments.

I do not have many minor comments but several as below:

Line14, CO2 should be corrected;

Corrected.

Line 17, Sphagnum mosses are ok;

We have removed the word peat.

Line 20, subgenera but not genera;

Thank you we have corrected this.

Line 20 and 21, along the gradient of ... would be fine;

Corrected.

Line 211, I am not sure which term is more suitable between moss core and moss monolith?

If the reviewer does not have a strong preference, we prefer the term core as the derivation of monolith (one stone), suggests a rigid hard object.

Line 244, it would be better if the unit for a, b and d can be changed as cm3, cm2 and cm3; 

Thank you for the suggestion, we have now changed it.

Line 498 to 581, the scientific name for plants should be italic.  

Corrected.

Reviewer 3 Report

Comments and Suggestions for Authors

In this study, the authors present new data on the role of Sphagnum morphological traits and colony density in water storage capabilities. Four different Sphagnum species (S. fallax, S. capillifolium, S. papillosum and S. inundatum) that occupy different positions along the water table were used to investigate desiccation avoidance strategies that vary among species and habitats. As reduced groundwater is one of the main threats to peatland functioning, the effect of desiccation on Sphagnum species is of particular interest for habitat restoration. The manuscript is well-written, the results are clearly presented and properly discussed. In my opinion, the manuscript could be accepted for publication in Plants in its current form.

Author Response

Thank you for revising out manuscript and considering it well written and clearly presented.